# Compilation of Evidence Supporting the Role of a T Helper 2 Reaction in the Pathogenesis of Acute Appendicitis

**DOI:** 10.3390/ijms25084216

**Published:** 2024-04-11

**Authors:** Nuno Carvalho, Ana Lúcia Barreira, Susana Henriques, Margarida Ferreira, Carlos Cardoso, Carlos Luz, Paulo Matos Costa

**Affiliations:** 1Serviço Cirurgia Geral, Hospital Garcia de Orta, 2805-267 Almada, Portugal; analuciapb19@gmail.com (A.L.B.); susanahenriques@campus.ul.pt (S.H.); margarida.s.f@gmail.com (M.F.); carlosluzcorreia@gmail.com (C.L.); paulomatoscosta@gmail.com (P.M.C.); 2Faculdade Medicina, Universidade Lisboa, 1649-028 Lisboa, Portugal; 3Dr. Joaquim Chaves, Laboratório de Análises Clínicas, 1495-068 Algés, Portugal; carlos.cardoso@jcs.pt

**Keywords:** allergy, appendicitis, appendicular lavage fluid, eosinophils, hypersensitivity type I reaction, mast cells

## Abstract

Despite being the most common abdominal surgical emergency, the cause of acute appendicitis (AA) remains unclear, since in recent decades little progress has been made regarding its etiology. Obstruction of the appendicular lumen has been traditionally presented as the initial event of AA; however, this is often the exception rather than the rule, as experimental data suggest that obstruction is not an important causal factor in AA, despite possibly occurring as a consequence of the inflammatory process. Type I hypersensitivity reaction has been extensively studied, involving Th2 lymphocytes, and cytokines such as IL-4, IL-5, IL-9 and IL-13, which have well-defined functions, such as a positive-feedback effect on Th0 for differentiating into Th2 cells, recruitment of eosinophils and the release of eosinophilic proteins and the production of IgE with the activation of mast cells, with the release of proteins from their granules. Cytotoxic activity and tissue damage will be responsible for the clinical manifestation of the allergy. AA histological features are similar to those found in allergic reactions like asthma. The intestine has all the components for an allergic immune response. It has contact with hundreds of antigens daily, most of them harmless, but some can potentially induce an allergic response. In recent years, researchers have been trying to assess if allergy is a component of AA, with their latest advances in the understanding of AA as a Th2 reaction shown by the authors of this article.

## 1. Introduction

### 1.1. Immune Defenses

In vertebrates, immune defenses consist of two subsystems—innate and adaptive [1,2]. The innate immune system uses phagocytic cells to control antigens, while the adaptive system responds specifically to each antigen, through cellular and humoral mediation mechanisms [3].

Immune system cells originate from pluripotent cells, which differentiate into two lines, the lymphoid line, with B cells, T cells and TNK cells, and the myeloid line with phagocytic cells, polymorphonuclear neutrophils (PMNs) and monocytes [4].

In the innate immune response, the effector mechanisms are diverse, including antimicrobial peptides, cytotoxic molecules, phagocytosis, and innate cytokines, such as Tumor Necrosis Factor (TNF)-α, Interleukin (IL)-6 and acute-phase proteins [5].

PMN and macrophages constitute the first line of defense against aggression via an immediate and non-specific response by using a repertoire of hereditary receptors to recognize a set of antigens [6].

PMNs are fundamental in the innate response; they envelop microorganisms, and produce cytokines and phagocytose being recruited to inflamed and infected tissue [6,7].

The infiltration of the *muscularis propria* of the appendix by PMNs is essential for histological confirmation of acute appendicitis (AA) [8].

Eosinophils are granulocytes, habitual residents of the intestinal mucosa, which participate in allergic reactions, where they release several mediators that are toxic for epithelial cells and smooth muscle [5,9].

The host response against either nematodes or allergies has common immunological mechanisms, such as IgE production, which is dependent on Th2 response, and activation of mast cells and eosinophils. This response can be protective against parasite aggression, but in allergies it is clearly associated with symptoms [10,11].

A hypersensitive reaction is an undesirable response of the adaptive immune system to harmless antigens, which sometimes even causes serious illness, due to tissue damage [12]. Any mucosa can be affected by allergies, and the reaction can be local, which means that it can manifest itself in the target organ, such as with allergic rhinitis and asthma, or it can be systemic, like in anaphylaxis [13,14].

The presence of an IgE-mediated response to eosinophils defines an allergy which results in target-organ and systemic inflammation [9,12].

A type I hypersensitivity reaction classified according to Gell and Coombs, can be summarized [9,13,15] (Figure 1): upon first contact with the antigen, the antigen will be presented by the antigen-presenting cell (APC) to naïve lymphocytes (Th0), which will differentiate into CD4+ lymphocytes, more specifically T-helper 2 (Th2) lymphocytes. 

IL-4, IL-5, IL-9 and IL-13 are produced by Th2 cells, in a type 2 immune response [16,17]. IL-4 has a positive feedback effect on Th2 cells and induces the differentiation of B cells into IgE-producing cells [17,18]. IL-5 interferes with all aspects of eosinophil activity, being called a chemotactic factor for eosinophils [17,18,19]. IL-9 interferes with the activity of eosinophils, mast cells and lymphocytes [18]. IL-13 has overlapping activity with IL-4 [17,20,21]. Lastly, the IL-4 and IL-13 cytokines promote smooth-muscle contraction [22].

B cells activated by IL-4 and IL-13 will produce the corresponding IgE, which will attach itself to high-affinity receptors on the surface of mast cells and basophils. 

In a subsequent contact with the antigen, mast-cell degranulation will occur, releasing histamine, prostaglandins and leukotrienes, which are responsible for allergic manifestations [23].

Histamine is the main mediator of acute inflammation and immediate hypersensitivity [24]. The deleterious effects of allergic reactions are caused by histamine, since it increases the contraction of smooth muscle, induces vessel dilation, increased vascular permeability, neutrophil and eosinophil chemotaxis [25,26] (Figure 1).

Serotonin intervenes in allergic processes by inducing smooth-muscle contraction and mucus production [27].

Allergic immune response is complex event, influenced by several factors, namely genetic background, exposure route, dosage, and structural characteristics of the allergen [13].

The allergic reaction is thus divided into two phases: the first one being sensitization and memory upon first contact with the antigen and the second one, which is the effector phase, happening when there is another contact with the antigen [28].

Allergy can also be associated with insufficiency of Treg and Th1 cells [29]. In an experimental model of chronic asthma, spontaneous resolution was shown to correlate with the accumulation of Treg cells [30]. Tregs produce anti-inflammatory cytokines by means such as IL-10, which has a suppressive effect on other cells of the immune system, such as B cells, T cells, eosinophils, or mast cells, suppressing the inflammatory process [29,30].

The hypersensitivity reaction aims to protect the organism, isolating it from potential antigens with a pathogenic effect; it can, however, be inadequate and responsible for allergic disease, such as allergic rhinitis, asthma, atopic dermatitis, or food allergies, which are a public health problem, affecting more than 25% of the population [11,31,32].

An historical perspective suggests that many defensive mechanisms are excessive or entirely unnecessary in the current environment, but that they may have made sense in an evolutionary context.

### 1.2. Appendicitis

The term ‘appendicitis’ was introduced in 1886 by Reginald Fitz, a Harvard pathologist [33]. 

AA is the most frequent abdominal emergency-surgery pathology, having an estimated lifetime risk of 6–7% [34] and being the most common diagnosis of both children and adults which requires hospitalization [35].

It is curious that, for such a common pathology, we regrettably know little about its etiology, physiopathology and treatment.

For many years, the standard treatment for AA was appendectomy, which is now controversial, as in particular clinical cases, like APA, conservative treatment is proposed, instead of appendectomy [36].

The common teaching is that AA is caused by appendix luminal obstruction [8]. However, this is seldom seen in surgical clinical practice [37], where obstruction seems to be the consequence rather than the cause of AA [38].

The appendix is a safe house for bacteria [39]. Bacteria are always present in the appendix and their contribution to AA is disputed. Some authors report that AA is an infectious disease, but it seems that bacteria are more bystanders rather than an active contributor to the disease.

It seems that appendicular invasion is secondary to obstruction or inflammation of the appendix, and not the primary event [40,41].

Many studies of microbiota and appendicitis have only shown associations, not definitive causal relationships between microbiota and AA [42].

Conservative treatment of AA without antibiotics has been recently performed, thus suggesting a non-bacterial etiology of AA [41].

Tsuji was the first author to propose the occurrence of an immune process in AA. In fact, he noted that the lamina propria of the appendicular wall in cases of AA was infiltrated by lymphocytes. He postulated that an non-identified antigen could stimulate their presence at the appendicular wall, inducing inflammation-AA [43,44]. 

Later, Aravindan, when studying AA specimens, noted the common presence of eosinophil infiltration in the tunica muscularis, intramuscular edema and mast-cell degranulation. He proposed that AA was initiated by a type I hypersensitivity reaction, as the histological features are similar to those found in type I hypersensitivity [45].

The conception is tempting. The appendix is predominantly a lymphoid organ, having in its mucosa all the cells that are involved in allergy, like CD4+, CD8+ effector cells and regulatory T cells, B cells, macrophages, dendritic cells and eosinophils [46].

The digestive tract is one of the main antigen entrances in the body. Most of them are tolerated by the organism, but some can elicit an allergic reaction. By analogy with bronchospasm in asthma, luminal appendicular obliteration can be provoked by muscular-wall appendicular contraction, in response to an antigen, and the sequence of events can lead to ischemia, gangrene and necrosis [47,48]. Any segment of the intestine can be involved in this process, but the appendix is more susceptible, because its smaller caliber allows for the obstruction to more easily occur [45].

Several recent studies have reinforced the hypothesis that there is a Th2 immune reaction in AA [19,45,47,49,50,51,52,53,54,55,56,57,58,59] (Table 1).

## 2. T-Lymphocytes: IL-4, IL-5, IL-9 and IL-13 

Cytokines are proteins produced by cells that alter the behavior of other cells, affecting almost all biological processes [5,60]. Only red blood cells do not produce and respond to cytokines [60].

The main characteristics of cytokines are the multiplicity of their biological activities and their functional redundancy, as different cytokines can perform the same function [60,61]. 

B and T lymphocytes mediate the adaptive immune response, through the recognition of antigens [6,61,62]. 

T lymphocytes are produced in bone marrow, and at the thymus they are divided into CD8+and CD4+ T lymphocytes. The latter are called T helper (Th) cells [4,5,12].

CD4+ T lymphocytes will differentiate into Th1, Th2, Th17 or Treg, according to the pattern of cytokines detected in the environment, so as to regulate the immune response [11,61].

Th1 cells secrete Th1 cytokines, such as IL-2, TNF-α and Interferon-γ (IFN-γ), which activate macrophages and cytotoxic T cells during cell-mediated immunity [55,63,64].

Th2 cells secrete Th2 cytokines, such as IL-4, IL-5, IL-9 and IL-13, and stimulate B cells to produce antibodies, in particular IgE in the course of humoral-mediated immunity [61,65,66] (Figure 1).

Excessive Th1 response is associated with several autoimmune and inflammatory disorders, whereas increased production of Th2 cytokines is involved in atopic diseases like allergies and asthma [64,67].

Having bronchoalveolar lavage fluid as a reference, we developed the appendicular lavage fluid (ALF) concept for studying immuno-inflammatory changes in the appendix [47]. In summary, after appendectomy, a gauge was inserted in the cecal luminal surface of the appendix and 3 mL of saline 0.9% was instilled and collected for ALF. Instillation and recollection was performed 3 times [47]. 

The resulting effluent (ALF), after adequate preparation, can then be used for the determination of not only lymphocyte populations, but also of other immune cells and cytokines.

Th2 cytokines IL-4, IL-5, and IL-9 were evaluated in the ALF in three groups of patients: acute phlegmonous appendicitis (APA), acute gangrenous appendicitis (AGA), and the control group, which is comprised of patients that have a clinical diagnosis of AA and were submitted to appendectomy, but had a normal histology (negative appendicitis) [47].

Differences in cytokines were found between the three groups, and an especially significant difference was found between phlegmonous and negative appendicitis. We argue that ALF reflets the inflammatory environment of AA. Therefore, the Th2 cytokine profile in ALF is compatible with allergy [47].

A recent study in pediatric patients with AA evaluated the hypersensitivity type I inflammatory response via ALF, by assessing the levels of IgE, IL-4, IL-5, IL-6 and IL-9 [51]. The patients were divided into two groups: the infant group (children under 3 years old) and the pediatric group (children between the ages of 3 and 15) [51].

The infant group had the higher cytokine level, with the exception of the IL-5 levels, which were not significantly different. The authors thereby concluded that the inflammatory response in children with AA was associated with type I hypersensitivity and the response was more intense in the infant group [51]. Nonetheless, the study has a main limitation, which is the lack of negative appendectomy data for comparative analysis, together with the small size of the sampled data [51]. 

A prospective study analyzed serum concentrations of IL-4, IL-9, IL-13 and total IgE in 138 patients who were less than 15 years old, with AA [55].

Serum concentrations of IL-4 and IL-9 did not affect the risk of complicated appendicitis, but at the same time high levels of IL-13 were associated with an increased risk of complicated appendicitis [55].

A retrospective study evaluate a total of 140 pediatric patients with AA [52]. 

IgE, IL-4, IL-5, IL-6, IL-9 and IL-13 serum levels were collected. The group was divided into two: complicated AA (CAA) and uncomplicated AA (UAA). The authors concluded that IgE, IL-6, and IL-13 levels can be risk factors for CAA. The combination of these markers can thus be used to diagnose CAA [52]. 

We evaluated the Th2 responses in AA via the determination of key cytokines, in particular IL-4 and IL-13, at a local level, in the ALF, and at a systemic level, in the peripheral blood of 46 patients undergoing appendectomy for AA and 14 appendectomy specimens from patients undergoing right colectomy for oncological pathology (unpublished data) [68,69].

The study shows that serum IL-4 is elevated in AGA and APA. No difference was found between the two types of appendicitis (unpublished data). 

IL-13 concentrations were also higher for AGA and APA, both in serum and ALF, and again, no difference was found between the two types of appendicitis (unpublished data). A recent prospective study in children showed that high levels of IL-13 appear to be associated with an increased risk of complicated appendicitis [55].

In ALF, the highest concentrations of IL-4 and IL-13 were present in complicated appendicitis, perforated appendicitis and peritonitis, which are associated with more destructive forms of appendicitis (unpublished data). 

Conservative treatment of uncomplicated appendicitis is feasible and safe in the immediate term, but failures and recurrences are frequent. It is essential to identify patients with uncomplicated appendicitis, as they may be proposed for conservative therapy [70]. 

Nonetheless, the different cytokines that were studied had no discriminatory power with clinical relevance for the differential diagnosis of complicated versus uncomplicated AA.

Eosinophil tissue infiltration is common during the allergic reaction in the targeted organ, and is dependent on IL-5 serum levels.

We designed a prospective study in order to evaluate eosinophil and IL-5 serum levels, together with IL-5 ALF levels and appendicular-wall eosinophilic infiltration. The study included 65 patients with AA and 18 patients in the control group, as they were clinically diagnosed with AA, submitted to appendectomy, and had normal appendicular histological findings [19]. 

Appendicular-wall eosinophil infiltration was determined manually by counting eosinophils by light microscopy. Eosinophil infiltration was higher in APA than in AGA. In ALF, the higher levels of IL-5 were also observed in the APA group [19].

A positive correlation was found between the appendicular-wall eosinophilic infiltration and the IL-5 concentrations in both blood and ALF [19].

The authors concluded that a hypersensitivity type I allergic reaction is present in the target organ, the appendix, as eosinophils are involved in the pathogenesis of Th2 immune-mediated diseases and are present in the target organ of the allergy [19,71,72]. 

## 3. B-Lymphocytes IgA, IgE, IgG e IgM

B cells produce antibodies, activate complement and antibody-dependent cell-mediated cytotoxicity pathways contributing to adaptive immunity [5]. B cells differentiate into antibody-secreting plasma cells after exposure [4].

Immunoglobulins (Igs), or antibodies, are glycoproteins that attach to antigens, which will be destroyed by cells of the phagocytic system and by complement proteins. 

Mammals express five types of Igs: IgA, IgD, IgE, IgG and IgM.

IgA is the main isotype of mucosal immunity, which protects epithelia, whereas IgE evolved as a defense mechanism against parasites, but they are best known for their pathological effects in allergy [73,74]. IgM is the main immunoglobulin of the primary immune response.

B lymphocytes initially produce IgM, and under the influence of T lymphocytes they can shift towards the production of IgA, IgE or IgG [1,9].

The orchestration of B and T cell responses is an extremely complex process, since they cooperate with and influence each other [3].

Curiously, knowledge about the immune response in AA is scarce. Igs play a central role in the humoral immune response, providing clinically relevant information on the state of the humoral response [75].

Generic humoral immune response was evaluated in a prospective study in patients with APA and AGA and in the control group (patients with negative histology) [56].

Although the levels of IgA, IgE, IgG and IgM were higher in the appendicitis group, in comparison with the control group, no significant statistical difference was found between the two [56]. 

Das et al. showed that serum IgE levels were higher in patients with recurrent AA. The authors hypothesized that persistently high serum-IgE levels may lead to recurrent appendicitis, showing a similarity with bronchial asthma [75].

A cohort study of 605 children undergoing appendectomy showed that the risk of complicated appendicitis is lowered 3 times in those with IgE-mediated allergy when compared to those without allergy [49]. The explanation to this event can only be speculative [49].

A prospective study which included 138 pediatrics patients with AA did not show any IgE blood-level differences between patients with either complicated or un-complicated AA [55]. 

The serum levels of IgE can be normal in patients with allergies, as most of the IgE is located in the target organ of allergy, where it binds to mast cells. IgE presence in blood is a consequence of its excessive local production.

IgE is present in the nasal mucosa of patients with allergic rhinitis, but not in the control group [76,77]. Local allergic rhinitis describes a Th2-type of nasal mucosal inflammation, in which IgE antibodies are produced in the nasal mucosa, which is why the nasal allergen provocation test is positive and systemic atopy is not proven [78]. This can also happen in AA. 

In a retrospective study, where a monoclonal antibody against human IgE was used, we evaluated the presence of IgE in appendicular specimens of histologically confirmed 38APA, 27 AGA, and in the control group, which was composed of 52 incidental appendectomies and 17 negative appendicitis. The slides were visualized with light microscopy and a standard procedure was used to manually count the positive-IgE stained cells. The highest number of IgE-positive appendicular cells were present in APA (Figure 2) [50].

These data suggest that an allergic reaction is present in AA [50].

Levels of IgE were evaluated in ALF from patients who were under 14 years old with AA. IgE levels were higher in the group who were less than 3 years old, when compared to the group aged between 3 and 14 years old. The authors concluded that a type I hypersensitivity-induced inflammatory response is present in AA, besides being more intense in the infant group [51].

## 4. Eosinophils: Eosinophil-Derived Neurotoxin, Eosinophilic Cationic Protein, Eosinophilic Peroxidase 

Eosinophils play a fundamental role in the host’s defense against parasites, but also in the pathogenesis of Th2 immune-mediated disorders, and their blood counts are used for treating and monitoring eosinophil-related diseases, such as asthma, atopic dermatitis and allergic rhinitis [79]. 

Eosinophil activation induces the release of active substances from their granules. Eosinophil-derived neurotoxin (EDN), eosinophil cationic protein (ECP), eosinophil peroxidase (EP) and major basic protein (MBP) are among the most studied eosinophil proteins, and are markers of eosinophil activation [79,80,81,82].

ECP, EP and EDN granule proteins have been identified as potential biomarkers of type 2 eosinophilic disease [82].

Eosinophils are key players in allergic inflammation and tissue homeostasis in hypersensitivity diseases, as their biological products are toxic to cells and human tissues [83].

Eosinophil infiltration of the muscularis mucosa of appendicular specimens is a universal event in AA [45]. Eosinophil presence in *muscularis propria* is more sensitive than neutrophils for early acute clinical symptoms for appendicitis. Increased eosinophils in *muscularis propria* may be a marker for early symptomatic appendicitis [84]. Eosinopenia is common in sepsis, and infection can reduce the eosinophil peripheral blood count [85]. A retrospective study demonstrated a decrease in eosinophil counts from negative to APA, with the lowest number in AGA. [86]. Allergy and sepsis are confounding factors for the presence of eosinophils in peripheral blood. 

Eosinophilia is not common in AA, even in patient where AA is associated with parasitic infection, like *enterobius vermicularis* [87]. Parasites can present in AA, but no definitive causal relation has been established [87]. In the present series, no parasite was found in an appendectomy specimen. 

Eosinophils present in appendicular-wall infiltration of patients with APA, AGA and the control group were evaluated. The highest presence of eosinophils was observed on the phlegmonous appendicitis walls [19]. Eosinophilic infiltration is found at the target organ of allergy, like in nasal rhinitis, asthma, atopic dermatitis and eosinophilic esophagitis [72]. The data are compatible with a hypersensitivity type I allergic reaction in the target organ, the appendix [19].

If eosinophils are not just witnesses, then eosinophil granule proteins are supposed to be increased at local and systemic levels. The literature is scarce in evaluating eosinophil granule proteins and AA. 

A recent prospective study evaluated eosinophil granule protein concentrations in ALF and blood, thus showing that ECP and EP levels are strongly elevated in AA, in comparison with the control group [53]. The authors concluded that the data suggest the presence of an allergic component in AA, as these eosinophil granule proteins are involved in the Th2 immune response [53]. 

## 5. Mast Cells: Tryptase, Histamine, Serotonin

Mast cells s are key factors in the Th2 immune response by releasing inflammatory mediators from their granules upon stimulation by IgE-specific antigens [54]. Elevated serum tryptase concentrations have been observed in several diseases, such as systemic allergic reactions [88].

Mastocytes are strategically located, interfacing with the environment, such as the gastrointestinal tract, where they act like sentinels that recognize pathogens and initiate an immune response by releasing their granules [89,90,91]. However, this release can induce damage in normal tissue and become pathogenic [90].

Tryptase is the main protease stored within mast cells [92] and one of the most well-known mast cell granule mediators, besides being responsible for the clinical manifestations of allergies and playing a central role in IgE-mediated allergies [89,90].

Tryptase function has not been completely elucidated, but it is believed that it promotes pro-inflammatory actions [88].

A single prospective study simultaneously evaluated tryptase levels in blood and ALF in patients with AA and the control group, which was comprised of patients submitted to right colectomy for right-side colon cancer [54]. 

Appendectomy was performed after surgical removal of the right colon, having been followed by the ALF procedure. Tryptase levels were notoriously different among histological groups, both in ALF and serum. ALF tryptase levels were higher than serum levels, and these differences could reflect an intense inflammatory local reaction in the appendicular lumen. Tryptase concentrations in ALF were 109 times higher in APA and 114 times higher in AGA than in the control group [54]. 

There were differences between tryptase serum levels, but these were not as striking as those from ALF. In serum, tryptase levels were 6.5 times higher in APA and 11 times higher in AGA, when compared to the levels of the control group [54].

In the same study, no differences were found in ALF and serum for the levels of histamine and serotonin. The authors argue that due to their short half-lives, when the determination was made it was no longer present by the time the samples were collected. Another study showed that serotonin levels decreased with time during the evolution of clinical presentation [93]. No other studies have been published on this subject.

## 6. Discussion

Despite being the most common pathology in emergency surgery, the cause of AA remains a mystery [70]. The reason for this is that the obstructive component of the appendicular lumen, which is pointed out as the cause of AA, seems to be more a consequence than the cause of AA. Fecalith or hyperplasia of lymphoid tissue are the putative causes of obstruction [94]. 

Fecaliths are common in a population without appendicitis [95]. One study even showed a higher incidence of fecaliths in autopsied appendices than in appendectomy specimens associated with clinical AA (25% versus 3%) [96].

Likewise, lymphoid hyperplasia is a physiological process more prevalent in the non-inflamed appendix than in AA [97]. This is further corroborated by a study of 1711 appendicitis cases, where the presence of lymphoid tissue hyperplasia was only observed in 15 cases [98]. A virus can induce lymphoid hyperplasia, but a recent multicenter retrospective cohort study in children showed no correlation between AA and common viral pathogens [99]. 

It thus seems unlikely that obstruction of the appendicular lumen by either fecalith or lymphoid hyperplasia is the primary cause of AA. Luminal obstruction by foreign bodies is a well-known, but exceedingly rare, cause of AA [100]. 

Recent investigations were focused on identifying a hypersensitivity type I reaction in AA, due to epidemiologic, experimental and clinical data supporting the notion that a Th2 immune reaction can happen in AA [19,45,47,49,50,51,52,53,54,55,57,58,59] (Table 1).

In summary, it is extraordinary that for such a common and seemingly simple disease, little is known about its etiology and pathophysiology. 

Researchers are provided with a unique opportunity to investigate these questions, as AA is a disease which commonly allows for the inflamed organ to be completely removed and available for analysis [101]. A control group is made up of patients with a clinical picture compatible with AA who are undergoing appendectomy but whose histology is normal [102].

Therefore, even nowadays, the 1925 JAMA publisher’s statement that “Acute appendicitis is a disease frequent enough to justify an extremely clear understanding of its etiology, but it would be rash to assert that this has been achieved” is proven to be true [103].

## 7. Conclusions

The etiology of AA is still an open field for research. Several sources of evidence point to a hypersensitivity type I reaction in AA. In fact, the different steps involved in allergic reactions are documented in AA at both the cellular and molecular level.

In a time of conservative treatment of AA, new strategic therapies can be brought forward if the concept of AA as an allergy is verified. Further investigations will continue to be awaited, bringing questions and answers to the understanding of etiology and the treatment of AA.

## Figures and Tables

**Figure 1 ijms-25-04216-f001:**
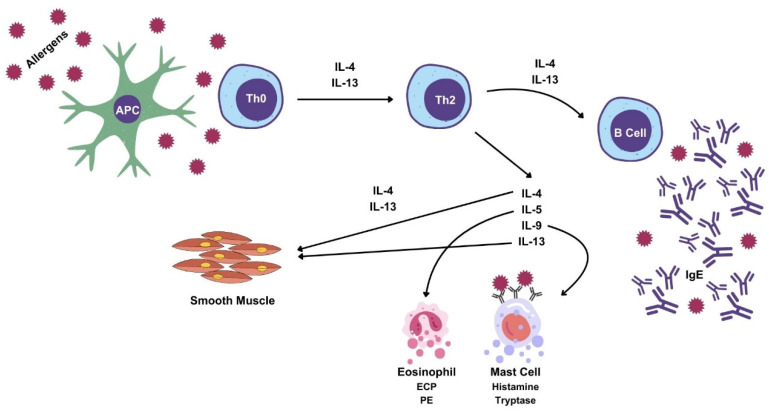
Type I hypersensitivity reaction: antigen-presenting cells (APC) present antigens to naive T cells (Th0). Th0 cells will differentiate into Th2 cells, which secrete IL-4, IL-5, IL-9 and IL-13. IL-4 has a positive feedback effect in Th0 cells, because it induces their differentiation into Th2 cells and also influences the differentiation of B cells into IgE-producing cells. The IgE then attaches itself to high-affinity receptors on mastocytes, with antigen binding and crosslinking of IgE antibodies inducing mast-cell granule protein release as a result. IL-5 promotes eosinophil activation, tissue eosinophilia and the release of eosinophil granule proteins, like the Eosinophilic Cationic Protein (ECP) and Peroxidase Eosinophilic (PE). IL-9 is involved in mast-cell proliferation, whereas IL-4 and IL-13 cause smooth-muscle hyperreactivity. IL-13 has similar biological activity to IL-4 (Courtesy of Sofia Guimarães, MD).

**Figure 2 ijms-25-04216-f002:**
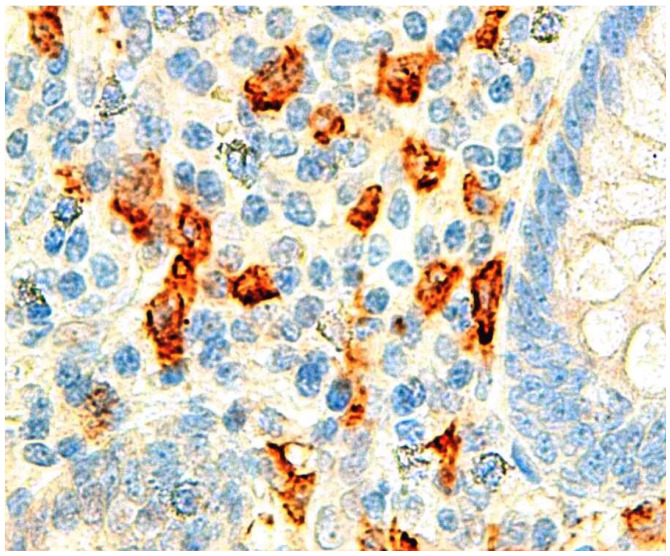
IgE immunostaining, acute appendicitis (Courtesy of Hélder Coelho, MD).

**Table 1 ijms-25-04216-t001:** Th2 immune response in Acute Appendicitis.

Component Evaluated	Reference	Study Design	Conclusion
Epidemiology	Salo M [49]	Cohort	IgE-mediated allergy protects from complicated AA
Epidemiology	Hasassri ME [58]	Case-control	Increased risk of AA in active asthma
Epidemiology	Kim SY [59]	Case-control	Increased risk of appendectomy in asthma
Appendicular wall eosinophils	Aravindan K [45]	Prospective	Eosinophils infiltrate appendicular wall in AA
Appendicular wall eosinophils	Carvalho N [19]	Retrospective	Eosinophils infiltrate appendicular wall in phlegmonous AA
Appendicular wall IgE	Carvalho N [50]	Retrospective	IgE positive cells higher in phlegmonous AA
Serum IgE	Das NM [57]	Prospective	IgE elevation in recurrent AA
Serum IgE	Carvalho N [56]	Prospective	No differences found between AA and control group
ALF IL-4, IL-5 and IL-9	Carvalho N [47]	Prospective	Th2 cytokine levels higher in AA
ALF IgE, IL-4, IL-5, IL-6 and IL-9	Pan ZB [51]	Prospective	IgE, cytokines elevated in AA
ALF/Serum IL-4 and IL-13	Unpublished	Prospective	Cytokines elevated in AA
Serum IgE, IL-4, IL-9, and IL-13	Gudjonsdottir J [55]	Prospective	IL-13 elevated in complicated AA
Serum IgE, IL-4, IL-5, IL-6, IL-9 and IL-13	Zhang T [52]	Retrospective	IgE, IL-6 and IL-13 elevated in complicated AA
ALF/Serum eosinophil granule proteins	Carvalho N [53]	Prospective	ECP and EP elevated in AA
ALF/Serum tryptase	Carvalho N [54]	Prospective	Tryptase levels elevated in AA

AA—Acute Appendicitis, ALF—Appendicular Lavage Fluid, ECP—Eosinophil Cationic Protein, EP—Eosinophilic Peroxidase.

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
