# Peer review of "Compilation of Evidence Supporting the Role of a T Helper 2 Reaction in the Pathogenesis of Acute Appendicitis"

_ijms, 2024, doi:10.3390/ijms25084216_

Round 1

Reviewer 1 Report

Comments and Suggestions for Authors

The authors present a narrative review of the evidence for Th2 reaction in the pathogenesis of acute appendicitis.

It is an appropriately structured article in this type of narrative review, in which no rigour is followed, unlike systematic reviews.

The introduction is excessively long, with short sentences that make it difficult to follow the logical thinking of the manuscript. I recommend shortening it and writing it in paragraphs, not in separate lines.

Throughout the manuscript it is stated several times that conservative treatment of acute appendicitis, even without antibiotics, is possible. This is an interesting reflection, because none of the authors are paediatric surgeons, who are the ones who face this reality every day. It is difficult to justify to parents the non-use of antibiotics in a patient with appendicular peritonitis... even if it appears to be an allergic reaction. However, all available evidence has been obtained from patients in whom the appendix has been removed.... my concern is the following: if the final result is the same (appendectomy), how is all this research justified?

The analytical pattern of the haemogram of these patients is absolutely infectious (neutrophilia with lymphopenia...), and in some patients there is eosinophilia, especially in patients with appendicitis secondary to helminth infection. Infection, that word that seems to be displaced by "allergy". I miss commenting on this as well.

The tables and figures in the manuscript are adequate.

Comments on the Quality of English Language

Moderate editing of English language required. I recommend proofreading the manuscript by a native speaker to correct syntactic and grammatical errors throughout the manuscript.

Author Response

Dear Reviewer 1

I am very grateful for the comments and suggestions that were introduced into the text, I hope that it is in line with your intentions, and that it can be recommended for publication.

The introduction was reformulated, reducing the content and condensing the separate lines into paragraphs.

References were added in relation to appendicular infection, helminths and considerations about the blood count, in particular eosinophils.

Most patients with acute appendicitis in adulthood undergo surgery. If the presence of an allergic component is confirmed, new therapeutic perspectives in acute appendicitis may open up. Until recently, rectal cancer was indicated for surgery; currently, a watch and wait policy is implemented, with firm scientific support. Could the same happen with acute appendicitis?

Best Regards

Reviewer 2 Report

Comments and Suggestions for Authors

In their review, the authors present an overview of the recent literature supporting the notion that a Th 2 immune response may occur in the pathogenesis of acute appendicitis. The review is interesting, well conceived and written, and relevant. The literature is up to date. My comments and suggestions for improvement are as follows:

1. The abstract is too short and not informative. There is no reference to the main topic of this review. The authors should add a few lines about the Th 2 response in the pathogenesis of acute appendicitis as this is a main topic of this article.

2. Paragraph 1.2 – lines 125 – 127 The authors state that the conventional wisdom is that AA is caused by luminal obstruction of the appendix (8). However, in clinical surgical practice this is rarely the case (37), as the obstruction appears to be the consequence rather than the cause of AA (38). This is questionable. As a surgeon, I see many cases with obstruction and almost the same proportion without obstruction. These cases without obstruction usually have enlarged lymph nodes that can obstruct the lumen from the outside, so the obstruction may be temporary. If obstruction of the lumen is not a factor, I would like the authors to comment and add a few lines in reference to a recent publication in which obstruction with foreign bodies favoured appendicitis (10.3390/children10010108). Perhaps the authors are right, but this constant is still speculation and should be restated. In addition, obstructive causes (as in this example) should be mentioned.

3. Paragraph 1.2 – lines 156 – 158 ''Any segment of the intestine can be involved in this process, but the appendix is more vulnerable, because its small calibre allows for the obstruction to more easily occur(Aravindan 1997). Please enter the reference number instead of ''Aravindan 1997''.

4. Paragraph 4 – lines 344 – 345 The authors state that eosinophils play a key role in allergic inflammation and tissue homeostasis in hypersensitivity diseases. I agree with this. Please add a few lines about the parasitic infestation of the appendix. Please also comment on the fact that in the majority of cases there is no eosinophilia (osinopils are present but no eosinophilia in blood) and only in very few cases there is an eosinophilic infiltrate in appendiceal wall (doi: 10.1016/j.jpedsurg.2021.09.038)

5. I would recommend that the authors add a short paragraph on viral infections as a cause of acute appendicitis. Several recent reports suggest a viral pathogenesis of acute appendicitis in some cases (doi: 10.3390/children10121912).

Comments on the Quality of English Language

Minor editing is requred.

Author Response

Dear Reviewer 2

I am very grateful for the comments and suggestions that were introduced into the text, I hope that it is in line with your intentions, and that it can be recommended for publication.

  1. The summary has been expanded to incorporate the Th2 response.
  2. Luminal obstruction was added with the proposed reference
  3. Aravindan reference has been updated
  4. Parasitic infection of the appendix was commented on, with the proposed reference
  5. The viral etiology was added, along with the proposed reference

Best Regards

Round 2

Reviewer 1 Report

Comments and Suggestions for Authors

The authors have adequately answered the questions raised by the reviewers.

Comments on the Quality of English Language

Minor editing of English language required